# Digital Quantum Simulation and Circuit Learning for the Generation of Coherent States

**DOI:** 10.3390/e24111529

**Published:** 2022-10-25

**Authors:** Ruilin Liu, Sebastián V. Romero, Izaskun Oregi, Eneko Osaba, Esther Villar-Rodriguez, Yue Ban

**Affiliations:** 1School of Materials Science and Engineering, Shanghai University, Shanghai 200444, China; 2TECNALIA, Basque Research and Technology Alliance (BRTA), 48160 Derio, Spain; 3Faculty of New Interactive Technologies, Universidad EUNEIZ, 01013 Vitoria-Gasteiz, Spain

**Keywords:** digital quantum simulation, quantum circuit learning, coherent state

## Abstract

Coherent states, known as displaced vacuum states, play an important role in quantum information processing, quantum machine learning, and quantum optics. In this article, two ways to digitally prepare coherent states in quantum circuits are introduced. First, we construct the displacement operator by decomposing it into Pauli matrices via ladder operators, i.e., creation and annihilation operators. The high fidelity of the digitally generated coherent states is verified compared with the Poissonian distribution in Fock space. Secondly, by using Variational Quantum Algorithms, we choose different ansatzes to generate coherent states. The quantum resources—such as numbers of quantum gates, layers and iterations—are analyzed for quantum circuit learning. The simulation results show that quantum circuit learning can provide high fidelity on learning coherent states by choosing appropriate ansatzes.

## 1. Introduction

The exponentially increasing scaling of degrees of freedom poses challenges in the computation for quantum chemistry [1]. Recent developments in quantum computing present new routes for the exploration of quantum chemistry by taking advantage of quantum resources and manipulating the states of matter. In the Noisy Intermediate-Scale Quantum (NISQ) era [2], seeking appropriate algorithms [3] is crucial for quantum dynamics digital simulation, which is one of the most promising applications.

As the first step of any quantum algorithm, the preparation of an initial state directly decides its success. Furthermore, in the Hamiltonian simulation, an efficient and accurate implementation of $U=e^{-iHt}$ is another crucial task. While the Hamiltonian of a molecule can be expressed easily in terms of its first quantization in real space [4] and simulate the kinetic part using the Quantum Fourier Transformation [5], the basis set for expressing the quantum states is smaller and more explicit in the second quantization [6], since a Fock state can be easily represented in the computational basis.

Analyzing and improving asymptotic scaling of resources on a quantum computer have been widely studied during the last decade [3,7,8,9]. In the NISQ era, low-depth circuits and reduced number of quantum gates are required to execute quantum algorithms in the presence of limited coherence time. Variational Quantum algorithms (VQA) [10,11,12], which are hybrid quantum-classical methods that take advantage from quantum and classical computation, have demonstrated to be resource-efficient strategies. Among them, Variational Quantum Eigensolvers (VQE) [13,14] can prepare the initial state and estimate the ground-state energy in a flexible and efficient way by choosing a suitable ansatz [15,16,17] in a parameterized quantum circuit (PQC). In addition, optimization in PQCs aims at achieving high fidelity of digital simulation by mitigating quantum errors [18]. To this end, gradient-descent based quantum circuit learning (QCL) [19] can be used to optimize the control function via finding out appropriate variational parameters.

Coherent states are a very special set of states forming the basis of continuous variables in quantum information [20,21], being useful in a wide variety of applications—for instance, to represent thermal and Schrödinger cat states, in the Mach–Zehnder interferometers configuration [22,23], in quantum metrology [24,25], in quantum cryptography [26], in quantum machine learning [27,28], among others. Simulation on coherent states in quantum circuits provides a fundamental and digital alternative to design and optimize quantum phenomena, quantum dynamics in gate-based quantum computers. Since the scalability of quantum algorithms is still limited by the complexity of quantum circuits, high-fidelity and efficient ways for digital quantum simulation of coherent states are indispensable for quantum technologies. In this paper, we propose a new method of digitized quantum simulation to simulate and optimize the generation of coherent states. We first use the creation and annihilation operators of the quantum harmonic oscillator to construct the displacement operator in the basis of a Fock state. By mapping the displacement operator acting on the vacuum state in the circuit, we can obtain coherent states with high fidelity. At the same time, to reduce the consumption of quantum resources in the digitized quantum simulation process, we propose VQA by combining the quantum circuit with a classical optimizer.

The paper is organized as follows. In Section 2, we introduce the coherent state and its digital simulation in a quantum circuit by trotterizing the displacement operator, where its decomposition with *N* qubits is generalized via the creation and annihilation operators. In Section 3, the Hardware Efficient Ansatz and Checkerboard Ansatz [29] are used to learn the coherent state. The quantum resources such as the number of quantum gates and iteration times are compared in different schemes. Finally, we give the Conclusion in Section 4.

## 2. Coherent State and Its Digital Simulation

First consider a quantum harmonic oscillator with the time-independent frequency ω, whose Hamiltonian can be written as (m=ℏ=1 in dimensionless units, where *m* is mass and *ℏ* is the reduced Planck constant): (1)$$H=\omega \left({\hat{a}}^{†}\hat{a}+\frac{1}{2}\right).$$

In the basis of a Fock space {|n〉}, the creation operator ${\hat{a}}^{†}$ and the annihilation operator $\hat{a}$ are written respectively as
(2)$${\hat{a}}^{†}=\left[\begin{array}{cccccc} 0 & 0 & 0 & \cdots & 0 & \cdots \\ \sqrt{1} & 0 & 0 & \cdots & 0 & \cdots \\ 0 & \sqrt{2} & 0 & \cdots & 0 & \cdots \\ \vdots & \vdots & \ddots & \ddots & \cdots & \cdots \\ 0 & 0 & \cdots & \sqrt{n} & 0 & \cdots \\ \vdots & \vdots & \vdots & \vdots & \ddots & \ddots \end{array}\right],\;\hat{a}=\left[\begin{array}{cccccc} 0 & \sqrt{1} & 0 & \cdots & 0 & \cdots \\ 0 & 0 & \sqrt{2} & \cdots & 0 & \cdots \\ 0 & 0 & 0 & \ddots & \vdots & \cdots \\ \vdots & \vdots & \vdots & \ddots & \sqrt{n} & \cdots \\ 0 & 0 & 0 & \cdots & 0 & \ddots \\ \vdots & \vdots & \vdots & \vdots & \vdots & \ddots \end{array}\right].$$

Using the displacement operator $\hat{D}\left(\alpha \right)=e^{\alpha {\hat{a}}^{†}-\alpha^{*}\hat{a}}$ to act on the vacuum state |0〉, we can generate the coherent state $|\alpha \rangle =\hat{D}\left(\alpha \right)|0\rangle$, which is also the eigenstate of the annihilation operator $\hat{a}$, satisfying $\hat{a}|\alpha \rangle =\alpha |\alpha \rangle$ with the arbitrary complex number α. As in Fock space *N* qubits map $n=2^{N}$ number of states, we directly express the coherent states in the basis of Fock states {|n〉}. It is feasible to simulate the generation of coherent states digitally in a quantum circuit, since the displacement operator $\hat{D}\left(\alpha \right)$ is a unitary operator.

Without loss of generality, we consider α to be a complex number a+bi, where *a* and *b* are real numbers and *i* is the imaginary unit, so that the displacement operator is written as
(3)$$\begin{array}{c} \hat{D}\left(\alpha \right)=e^{\left(a+bi\right){\hat{a}}^{†}-\left(a-bi\right)\hat{a}}. \end{array}$$

Here, we use the Hermitian matrix $Z_{1}=i\left(a^{†}-a\right)$ and the symmetric one $Z_{2}=-\left(\hat{a}+{\hat{a}}^{†}\right)$, where
(4)$$Z_{k}=i^{k}\left[\begin{array}{ccccc} 0 & {\left(-1\right)}^{k}\sqrt{1} & 0 & \cdots & 0\\ \sqrt{1} & 0 & {\left(-1\right)}^{k}\sqrt{2} & \cdots & 0\\ 0 & \sqrt{2} & \ddots & \ddots & \vdots \\ \vdots & \vdots & \ddots & \vdots & {\left(-1\right)}^{k}\sqrt{n+1}\\ 0 & 0 & \cdots & \sqrt{n+1} & 0 \end{array}\right],\;k\in \left\{1,2\right\},$$
is truncated from the subspace-embracing *N* qubits. For *N* qubits, we need $N2^{N-1}$ Pauli strings for decomposing $Z_{1}$ and $Z_{2}$ by using $\sigma_{\left\{0,1\right\}}^{D}\in \left\{I,\sigma_{z}\right\}$ (diagonal Pauli matrix) and $\sigma_{\left\{0,1\right\}}^{S}\in \left\{\sigma_{x},\sigma_{y}\right\}$ (skew-diagonal Pauli matrix). To express the formula in a more compact way, we can rewrite $Z_{1}=\sum_{l=1}^{N2^{N-1}}c_{l}A_{1}\left(l\right)$ and $Z_{2}=-\sum_{l=1}^{N2^{N-1}}c_{l}A_{2}\left(l\right)$, where $c_{l}$ and $A_{1,2}\left(l\right)$ are suitable definitions of each of the $N2^{N-1}$ constants and strings, respectively (see Appendix A for more details). By using the first-order Suzuki–Trotter formula [30,31], we can decompose the displacement operator into *M* number of Trotter steps, where
(5)$$\hat{D}\left(\alpha \right)\approx {\left(e^{-iaZ_{1}/M}e^{-ibZ_{2}/M}\right)}^{M}={\left[\left(\prod_{l=1}^{N2^{N-1}}e^{-iac_{l}A_{1}\left(l\right)/M}\right)\left(\prod_{l=1}^{N2^{N-1}}e^{ibc_{l}A_{2}\left(l\right)/M}\right)\right]}^{M}.$$

In fact, $\hat{D}\left(a\right)$ and $\hat{D}\left(ib\right)$ in Equation (Equation 3), representing the real and the imaginary displacements of the coherent state, respectively, are implemented in quantum circuits in completely different ways due to their distinct decomposition into tensor products of the matrices from $\sigma^{D}$ and $\sigma^{S}$. We use Qiskit [32] to implement the quantum circuit. Initialized in the vacuum states, i.e., |0〉 in all qubits, the quantum circuit generates coherent states after the implementation of the displacement operator (Equation (Equation 5)) into a finite number of segmentations. For instance, a two-qubit system (N=2) with four Fock states expressed in the computational basis, leads to the following real and imaginary parts for the displacement operator:(6)$$\begin{array}{cc} \; & \hat{D}\left(a\right)=e^{-iaZ_{1}}=e^{-ia[\frac{1+\sqrt{3}}{2}\sigma_{y}^{1}+\frac{1-\sqrt{3}}{2}\left(\sigma_{z}^{0}⨂\sigma_{y}^{1}\right)+\frac{\sqrt{2}}{2}\left(\sigma_{y}^{0}⨂\sigma_{x}^{1}-\sigma_{x}^{0}⨂\sigma_{y}^{1}\right)]}, \end{array}$$(7)$$\begin{array}{cc} \; & \hat{D}\left(ib\right)=e^{-ibZ_{2}}=e^{ib[\frac{1+\sqrt{3}}{2}\sigma_{x}^{1}+\frac{1-\sqrt{3}}{2}\left(\sigma_{z}^{0}⨂\sigma_{x}^{1}\right)+\frac{\sqrt{2}}{2}\left(\sigma_{x}^{0}⨂\sigma_{x}^{1}-\sigma_{y}^{0}⨂\sigma_{y}^{1}\right)]}. \end{array}$$

For example, the digital implementation of the displacement operator with real and imaginary displacements $\hat{D}\left(a\right)$ and $\hat{D}\left(ib\right)$ in a single Trotter step, where the total Trotter number is *M*, expressed by two qubits is shown in Figure 1.

To analyze the accuracy of the coherent state we derive, we define its fidelity
(8)$$F=|\langle \psi_{f}|\psi_{\mathrm{tar}}\rangle {|}^{2}$$
where $|\psi_{f}\rangle$ is the final state prepared by the circuit and $|\psi_{\mathrm{tar}}\rangle$ is the target state which can be calculated in Fock space as
(9)$$|\psi_{\mathrm{tar}}\rangle =|\alpha \rangle =\exp\left(-\frac{1}{2}{\left|\alpha \right|}^{2}\right)\sum_{k=0}^{\infty }\frac{\alpha^{k}}{\sqrt{k!}}|k\rangle .$$

Such a target has the probability of the *m*-th Fock state in the analytical form
(10)$$P_{m}={\left|\langle m|\psi_{\mathrm{tar}}\rangle \right|}^{2}=e^{-{\left|\alpha \right|}^{2}}\frac{{\left|\alpha \right|}^{2m}}{m!}=e^{-\langle n\rangle }\frac{{\langle n\rangle }^{m}}{m!},$$
which returns a Poissonian distribution centered at 〈n〉, with $\langle n\rangle =\langle {\hat{a}}^{†}\hat{a}\rangle ={\left|\alpha \right|}^{2}$.

We benchmark our method by digitally generating the coherent state |α〉=|1+i〉. Note that the coherent state is in bold in order to distinguish it from the Fock state |n〉. The circuit implementation is expressed in Figure 2a, where the blue (real displacement part) and red (imaginary displacement part) blocks act alternatively. Depending on the magnitude of |α| and meanwhile aiming at achieving the desired fidelity, one needs to choose the appropriate number of qubits to derive the coherent state with a desired fidelity in the circuit. Using 4 and 3 qubits gives rise to the fidelities over F=0.9999 with Trotter steps M≥14 and F=0.9986 with M≥20, respectively, as shown in Figure 2b. In Figure 2c, we plot the distribution of the coherent state |α〉=|1+i〉 prepared by digitally implementing the displacement operator $\hat{D}$ with 3 qubits and M=20 Trotter steps. It coincides well with the Poissonian probability distribution, indicated from Equation (Equation 10), where $\langle n\rangle =\langle {\hat{a}}^{†}\hat{a}\rangle ={\left|\alpha \right|}^{2}=2$. The analysis on the accuracy of the coherent state expressed in a truncated space with $2^{N}$ states, where *N* is the qubit number, can be found in Appendix B.

## 3. Coherent State Generation by Variational Quantum Algorithm

Gradient-based quantum circuit learning, a kind of VQA for supervised learning, aims at achieving the target state by lowering the cost function. In this section, we use a variational quantum circuit combined with classical optimizers to prepare coherent states. Inputting the ground state as the initial state into the circuit, processing the evolution in the form of three alternatives, we obtain the results at the output $|\psi_{f}\rangle$. To approach the target state $|\psi_{\mathrm{tar}}\rangle$ (Equation (Equation 9)) and by using gradient descent method based on the cost function
(11)$$C=1-|\langle \psi_{f}|\psi_{\mathrm{tar}}\rangle {|}^{2},$$
where $|\psi_{f}\rangle$ and $|\psi_{\mathrm{tar}}\rangle$ are the final state and target state as defined above, we find the optimized variational parameters in the ansatz. Such VQA is implemented in the Qiskit package where the optimizer is SLSQP from SciPy [33].

Essentially, a core component of a VQA is to find the appropriate ansatz which produces the coherent state with few gates and shallow depth in the circuit. Here, as shown in Figure 3, we apply two kinds of ansatzes by using 4 qubits: Hardware Efficient Ansatz (Figure 3a and Figure 3b, named as Scheme a and Scheme b, respectively) and Checkerboard Ansatz [29] (Figure 3c, named as Scheme c), since they are easy to be implemented in the current and near-term hardware. For the Hardware Efficient Ansatz which consists of a sequence of single-qubit $R_{x}$, $R_{z}$, $R_{x}$ gates and two-qubit-entangling Controlled-$R_{y}$ gates, we implemented two alternatives in order to see the relation between the fidelity and the number of control parameters. In Scheme a, the rotation angles of all the gates used in the circuit are the variational parameters with a number of 4N in each layer, where *N* is the number of applied qubits. While the initial values of all the angles are set to 1 empirically, their optimized values are derived from VQA. For Scheme b, the block of $R_{y}$ gates, with a number of *N* (equal to the qubit number) labeled in the block as shown in Scheme b, is repeated in each layer. In order to gain the desired fidelity, a number $M_{l}$ of repetition on each layer in the circuit is needed. For a circuit with $M_{l}$ layers, the number of control parameters in Scheme a and b are $4NM_{l}$ and $(3+M_{l})N$, respectively. The Checkerboard Ansatz, composed by N−1 blocks connecting neighboring qubits with a sequence of single-qubit $R_{x}$ and $R_{z}$ gates and two-qubit-entangling CNOT gates, is demonstrated in Scheme c, with a total amount of $5\left(N-1\right)M_{l}$ parameters.

The first two columns of Table 1 shows the number of the applied single-qubit and CNOT gates for three schemes in terms of the number of layers $M_{l}$ and the number of qubits *N*, where controlled-$R_{y}$ gates are decomposed into single-qubit and CNOT gates for comparison. In order to guarantee high-precision preparation of coherent states (F>0.9999), Schemes a, b, and c need 4, 6, and 6 layers for a 4-qubit system, respectively, as shown in Figure 4. Scheme b requires less quantum gates and controlled parameters even though it demands more layers.

For VQAs, the running time of the optimization is determined by the iteration steps, the number of parameters to be optimized, the initial value of the parameters, and the chosen optimizer. The last two columns of the Table 1 show the minimized iteration times and depth to guarantee high-precision preparation of coherent states (F>0.9999). The minimal steps needed to realize a high-precision preparation of coherent states |1+i〉 is achieved by Scheme b.

## 4. Conclusions

Coherent states and their digital quantum simulation are of significance in many fields, as a fundamental element in quantum computing, quantum machine leaning, and quantum optics. In this article, we proposed a new method for digitally simulating coherent states in quantum circuits. By expressing the Fock states with an appropriate number of qubits, we decomposed the displacement operator into Pauli matrices via the second quantization. A generalized formula on the Trotter expansion of the displacement operator with *N* qubits was demonstrated. The derived coherent states in quantum circuits coincided with a Poissonian distribution with high fidelity. Moreover, we also generated coherent states by gradient-based quantum circuit learning. Hardware Efficient Ansatz and Checkerboard Ansatz were used to find coherent states. Different schemes with distinct numbers of variational parameters were compared in terms of quantum resources and iteration times.

In the NISQ era, seeking efficient encoding methods, i.e., shorter circuit depth and less quantum gates while maintaining high fidelities, is always the object for the digital quantum simulation. Finding the optimized ansatzes to generate coherent states which can exhibit robustness in the presence of different kinds of noise will also be considered. These will be explored and addressed in our future work. Meanwhile, the transfer of digital quantum simulation from a coherent state to a squeezed state might be interesting to study. The quantum information processing hardware containing continuous-variable objects, such as mechanical or electromagnetic oscillators, instead of discrete-variable qubits, have demonstrated advantages in some aspects, such as quantum error corrections [34] and data encoding [35], although the physical implementation still demands further development [36]. We hope our method of digital simulation on coherent states will be useful for the dynamic simulation of quantum many-body systems and device design for quantum machine learning.

## Figures and Tables

**Figure 1 entropy-24-01529-f001:**
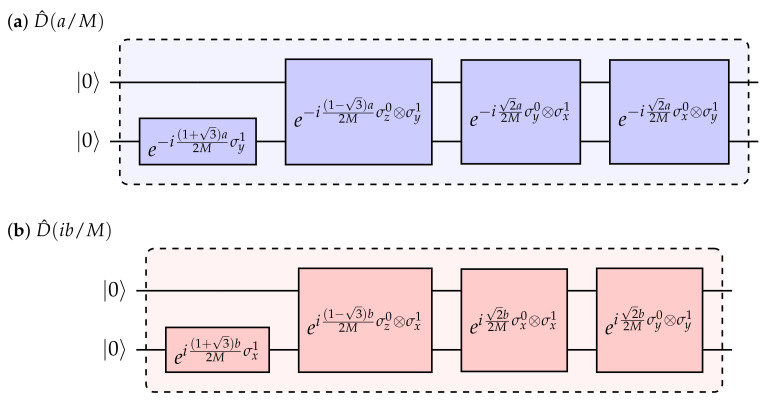
Diagram of a single step for a two-qubit circuit implementation of the displacement operator $\hat{D}$ with *M* Trotter steps. The displacement is (**a**) a real number *a* and (**b**) an imaginary value ib.

**Figure 2 entropy-24-01529-f002:**
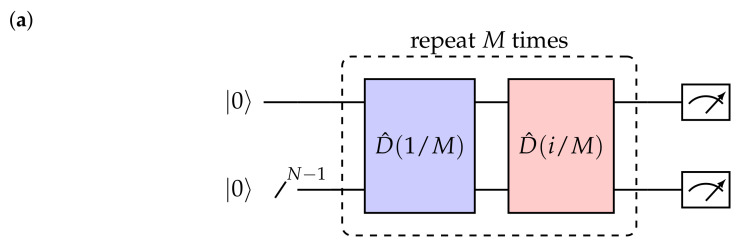
Preparation of the coherent state |α〉=|1+i〉 in a quantum circuit. (**a**) Circuit implementation where the blue and the red blocks represent the parts with real and imaginary displacements, respectively. (**b**) The dependence of the fidelity of the coherent state |1+i〉 prepared by 3 and 4 qubits on the number of Trotter steps *M*. (**c**) Fock distribution of the coherent state |1+i〉, simulated by 3 qubits and M=20 Trotter steps. The height of the yellow bar indicates the probability of finding particle in the *n*-th Fock state. The height in each Fock number coincides well with the Poissonian distribution (black-dotted) illustrated by $P_{m}$ (Equation (Equation 10)).

**Figure 3 entropy-24-01529-f003:**
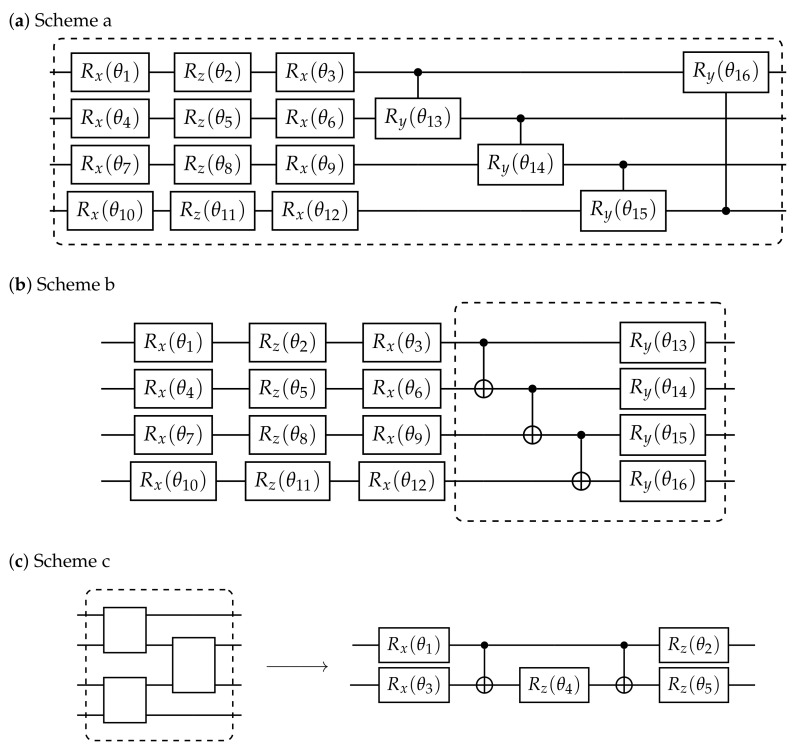
Three schemes of ansatzes adopted to do circuit learning for preparing coherent states by using 4 qubits, where the block (dashed line) in each subplot represents one layer in its corresponding circuit. Such a layer needs to be repeated in the circuit in order to obtain the desired fidelity. (**a**) Hardware Efficient Ansatz where the rotation angles of all the gates used in the circuit are the variational parameters. (**b**) Hardware Efficient Ansatz where the rotation angles of $R_{y}$ gates are set undetermined. (**c**) Checkerboard Ansatz whose unit of the neighboring two-qubit interaction with variational angles is depicted in detail on the right side.

**Figure 4 entropy-24-01529-f004:**
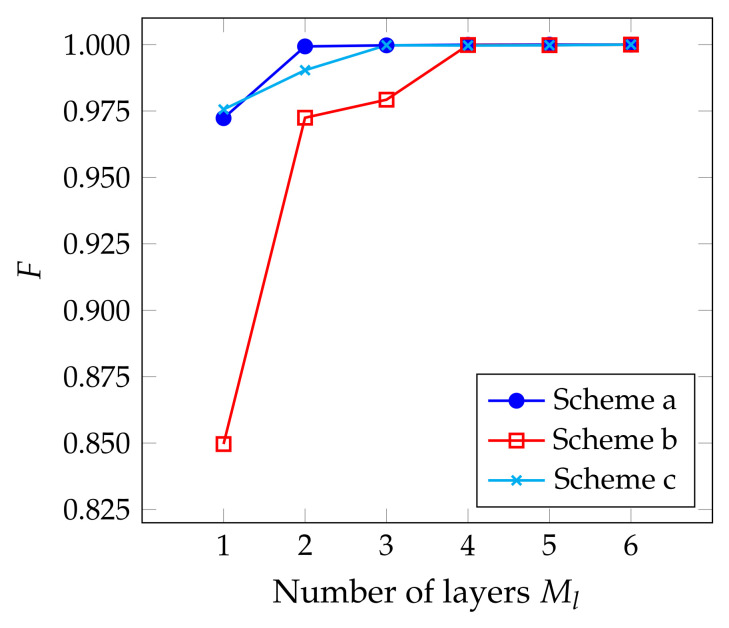
Dependence of fidelity *F* on the number of layers $M_{l}$ for preparing the target coherent state |1+i〉 with the application of the three schemes presented in Figure 3.

**Table 1 entropy-24-01529-t001:** Quantum gates used for the three schemes shown in Figure 3 and their corresponding results for acquiring F>0.9999.

Scheme	Number of Gates		Minimized Results for F>0.9999	
	Single-Qubit	CNOT	Iteration Times	Depth
a	$5NM_{l}$	$2NM_{l}$	4166	4
b	$(3+M_{l})N$	$\left(N-1\right)M_{l}$	2517	6
c	$5\left(N-1\right)M_{l}$	$2\left(N-1\right)M_{l}$	4099	6

## Data Availability

The data that support the findings of this study are available on request from the corresponding author.

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
