# Peer review of "Digital Quantum Simulation and Circuit Learning for the Generation of Coherent States"

_entropy, 2022, doi:10.3390/e24111529_

Round 1

Reviewer 1 Report

 In the present work authors investigate the possibility to simulate a 

coherent state using quantum circuits. 

In my opinion this paper is very interesting and the addressed subject is 

timely. Therefore in my opinion the it is suitable for publication 

on Entropy with only minor changes. 

Concerning them I have only few comments. 

i) The physical meaning of the parameter $m$ is unclear. I guess that it is the mass, 

but this should be better clarified. 

ii) The appearance of expressions such as $\sqrt{4}{2^{2}}$ is quite strange. I do not understand why the 

authors haven't simplified them.

Author Response

1: The physical meaning of the parameter $m$ is unclear. I guess that it is the mass, but this should be better clarified.

Response: We thank the Reviewer for reminding this point. To avoid confusion, we define $m$ is mass and set it to be $1$ in dimensionless units, which can be found above Eq. (1). In “Section 3, Coherent state generation by Variational Quantum Algorithm”, we define $M_l$ instead of $m$ as the number of the layers.

2: The appearance of expressions such as $\sqrt{4}{2^{2}}$ is quite strange. I do not understand why the authors haven't simplified them.

Response: We have simplified these expressions for Z_1 and Z_2 in Appendix A.

Reviewer 2 Report

1. A separate paragraph describing the structure of the essay is recommended at the end of the "Introduction".

2. Suggested additional analysis of evaluation criteria.

3. There is no mention of suggestions for future research directions in the conclusion.

4. The Summary and Conclusions suggest adding specific progression values.

5. Inconsistent formatting of papers, e.g. no line numbers on pages 3, 6, 8.

6. Coherent states are of great importance in quantum machine learning and additional references are suggested, “An Image Classification Algorithm Based on Hybrid Quantum Classical Convolutional Neural Network”, “Determination of quantum toric error correction code threshold using convolutional neural network decoders”.

Author Response

Response to the comments of Reviewer #2

1: A separate paragraph describing the structure of the essay is recommended at the end of the "Introduction".

Response: Following the Reviewer’s suggestion, we write a separate paragraph to describe the structure of the essay at the end of the “Introduction”.

2: Suggested additional analysis of evaluation criteria.

Response: To analyze the accuracy of the coherent state simulated in the quantum circuit or generated from circuit learning, we use the concept “fidelity” as the key evaluation criterion by comparing the final state with the ideal target state (the coherent state), seen in Eq. (8). We believe the value of the fidelity, which is commonly used in quantum information theory, can give the clear evidence to show the accuracy of generating coherent states by our method.

Meanwhile, the scheme using less quantum resource can be regarded as another evaluation criterion for implementing circuit learning. In the NISQ era, the circuit with less quantum gates and less layers is required to achieve high fidelities. From the point view of the hardware realization for two-qubit gates which still needs improvements, a scheme with reduced number of two-qubit gates is more favorable. Regarding this respect, in the Section “3. Coherent state generation by Variational Quantum Algorithm”, we compare quantum resources like number of layers, number of gates and depth of the circuit for the applied Hardware Efficient Ansatz and Checkerboard Ansatz.

3: There is no mention of suggestions for future research directions in the conclusion.

Response: In the NISQ era, shorter circuit depth and less quantum gates in the circuit to simulate coherent states are always the targets to be realized. In particular, we would like to find the optimized ansatzes which can provide the robust outputs of the circuit in the presence of different kinds of noise. We will take further exploration in this direction.

To clarify this point, we add the short discussion in the last paragraph of the Conclusion as “In the NISQ era, seeking efficient encoding methods, i.e., shorter circuit depth and less quantum gates while maintaining high fidelities, are always the objects for the digital quantum simulation. Finding out the optimized ansatzes to generate coherent states which can exhibit robustness in the presence of different kinds of noise will also be considered. These will be explored and addressed in our future work. Meanwhile, the transfer of digital quantum simulation from coherent state to squeezed state might be interesting to study.”

4: The Summary and Conclusions suggest adding specific progression values.

Response:

From the prospect of digital quantum simulation, the quantum information processing hardware containing continuous-variable objects, such as mechanical or electromagnetic oscillators, instead of discrete-variable qubits, have demonstrated advantages in some respects, such as quantum error corrections, although the physical implementation still demands further development.

To clarify this point, we add the sentence in the last paragraph of Conclusion as “The quantum information processing hardware containing continuous-variable objects, such as mechanical or electromagnetic oscillators, instead of discrete-variable qubits, have demonstrated advantages in some respects, such as quantum error corrections [34] and data encoding [35], although the physical implementation still demands further development [36]”.

5: Inconsistent formatting of papers, e.g. no line numbers on pages 3, 6, 8.

Response: In the new version of the manuscript, we have added line numbers on page 3 and 8. Since the text on page 6 is the caption of images, the line numbers are not added there.

6: Coherent states are of great importance in quantum machine learning and additional references are suggested, “An Image Classification Algorithm Based on Hybrid Quantum Classical Convolutional Neural Network”, “Determination of quantum toric error correction code threshold using convolutional neural network decoders”.

Response: It is really true as Reviewer mentioned that coherent states are of great importance in quantum machine learning. We have added two articles recommended by the Reviewer as Ref [27], [28] in the new version of the manuscript.